# Cascade Förster Resonance Energy Transfer Studies for Enhancement of Light Harvesting on Dye-Sensitized Solar Cells

**DOI:** 10.3390/nano12224085

**Published:** 2022-11-20

**Authors:** Mulugeta Tesema Efa, Jheng-Chang Huang, Toyoko Imae

**Affiliations:** 1Graduate Institute of Applied Science and Technology, National Taiwan University of Science and Technology, Taipei 10607, Taiwan; 2Department of Chemical Engineering, National Taiwan University of Science and Technology, Taipei 10607, Taiwan

**Keywords:** cascade Förster resonance energy transfer, light harvesting, dye-sensitized solar cell, zinc oxide, carbon dot

## Abstract

This work reports cascade Förster resonance energy transfer (FRET)-based n-type (ZnO) and p-type (NiO) dye-sensitized solar cells (DSSCs), discussing approaches to enhance their overall performance. Although DSSCs suffer from poorer performance than other solar cells, the use of composites with carbon dot (Cdot) can enhance the power conversion efficiency (PCE) of DSSCs. However, further improvements are demanded through molecular design to stimulate DSSCs. Here, a photosensitized system based on a cascade FRET was induced alongside the conventional photosensitizer dye (N719). To N719 in a DSSC is transferred the energy cascaded through donor fluorescence materials (pyrene, 3-acetyl-7-*N,N*-diethyl-coumarin or coumarin and acridine orange), and this process enhances the light-harvesting properties of the sensitizers in the DSSC across a broad region of the solar spectrum. PCE values of 10.7 and 11.3% were achieved for ZnO/Cdot and NiO/Cdot DSSCs, respectively. These high PCE values result from the energy transfer among multi-photosensitizers (cascade FRET fluorophores, N719, and Cdot). Moreover, Cdot can play a role in intensifying the adsorption of dyes and discouraging charge recombination on the semiconductor. The present results raise expectations that a significant improvement in photovoltaic performance can be attained of DSSCs exploiting the cascade FRET photonics phenomenon.

## 1. Introduction

Energy transfers between an energy-donating system (donor) and an energy-accepting system (acceptor) can be classified into two different types, radiative and non-radiative [1]. In the case of the former, a donor molecule transfers its photon energy to an acceptor molecule and, therefore, the distance between donor and acceptor is required to be close [1,2,3]. In the case of the latter, the excited energy of a donor molecule transfers its energy to an acceptor molecule through electron resonance, and, thus, this mechanism is called Förster resonance energy transfer (FRET). In FRET, the excited energy of a donor molecule gained by absorption of light energy, before returning to the ground state by emission, is transferred to a nearby acceptor molecule having slightly lower excitation energy by the Coulombic coupling mechanism.

Although dyes play a great role in the power conversion efficiency (PCE) of dye-sensitized solar cells (DSSCs) by harvesting the light, dyes absorbing in the near-infrared region to enhance the light-harvesting are difficult to synthesize. The strategy to fill this absorption gap is either co-deposition or multiple additions of photosensitizers with complementary absorption spectra. This is an important task for developing a new sensitizing dye system to enhance the light absorption ability and extend the spectral response range [4]. The FRET process [5] can occur intramolecularly or intermolecularly and enhance the light-harvesting of DSSCs. When a coumarin derivative (donor) was chemically bound to fluorescein (acceptor), this new molecule could perform intramolecular FRET [6]. When a rhodamine/dansyl fluorophore was bound via Cu^2+^ ion, this complex can carry out intermolecular FRET [7]. In intermolecular FRET, two different chromophores are involved in energy transfer. However, when the spectral overlap of the two chromophores is not sufficient, a third chromophore can be added to couple the energy transfer. This multi-step process is called cascade FRET [8].

In this work, the efficiency of energy transfer in cascade FRET was investigated in the three-fluorophore system of pyrene (Py), 3-acetyl-7-*N,N*-diethyl-coumarin (AC), and acridine orange (AO) and compared with the system of Py, coumarin6 (C6), and AO. The photovoltaic energy conversion efficiency of these systems was examined, and the photovoltaic parameters were elucidated and compared with those of a previously examined photovoltaic system consisting of ZnO or NiO, carbon dot (Cdot), and the di-tetrabutylammonium cis-bis(isothiocyanato)bis(2,2’-bipyridyl-4,4’-dicarboxylato)ruthenium(II) (N719) photosensitizer [9,10,11,12,13]. Finally, the energy transfer mechanism was discussed for this multi-fluorophore component system to enhance light harvesting. Thus, this investigation will give guidance for the future direction of DSSCs by improving light harvesting. It should also be noted that, although both ZnO@N719 DSSC and NiO@N719 DSSCs display only very low photovoltaic efficiency, their efficiencies increase when Cdot is added in these DSSCs [10,12]. Thus, a further increment of the PCE is expected after cascade FRET fluorophores are applied.

## 2. Experimental Procedures

### 2.1. Materials and Instruments

Py (98%), C6, and AO were purchased from Acros Organics (Waltham, MA, USA), Sigma Aldrich (St. Louis, MO, USA), and Chroma-Gesellschaft Schmid GmbH & Co., Ltd. (Stuttgart, Germany), respectively. AC has been synthesized and donated by Prof. K. M. Mahdevan and his co-workers at Kuvempu University, India. N719 was purchased from Solaronix (Darmstadt, Germany). Other reagents were commercial grade. Ultrapure water (18.2 MΩcm resistivity) was obtained from a Yamato Millipore WT100 (Tokyo, Japan).

ZnO, NiO, and their composites with Cdot (ZnO@Cdot and NiO@Cdot) are the samples previously synthesized and characterized [9,12,13]. White ZnO powder of 20.5 ± 9.5 nm size was prepared using a polyol method from a precursor of zinc nitrate hexahydrate. Black NiO powder, with short and long axes of 11.37 ± 0.53 and 16.48 ± 0.26 nm, respectively, was synthesized through a calcination method (at 350 °C) from a precursor of NiCl_2._ Cdot with 3 nm or less size in ZnO@Cdot and NiO@Cdot was prepared at a 2:1 mol ratio of ethylene diamine:citric acid.

AO was purified by column separation [14]. After AO was extracted by chloroform from an ethanol–water (59 *v*/*v*%) solution at pH 7, the chloroform portion was purified by passing through an aluminum oxide-packed column and by evaporating the solvent. The powder was solved in ethanol at pH 7 and dried. The purified AO was checked by using a thin-layer chromatography silica gel sheet (60 F254 25, Merck Millipore, Billerica, MA, USA).

Ultraviolet (UV)-visible absorption spectra (JASCO V-670, Tokyo, Japan) and photoluminescence (PL) spectra (F-7000, Hitachi High-Technologies Co., Tokyo, Japan) were measured for the characterization of the dye solutions.

### 2.2. DSSC Measurements

The paste of ZnO, ZnO@Cdot, NiO, or NiO@Cdot dispersed in isopropanol was spin-coated on an indium tin oxide (ITO) conducting glass plate precleaned in ethanol via sonication [10,12], and then sintered at 400 °C for 30 min. The prepared electrode was immersed for 5 h in a mixture of N719 dye (0.5 mM), fluorescence dyes (Py, AC, or C6 and AO) in ethanol at room temperature (~25 °C), rinsed with ethanol and water, and then dried.

Electrochemical measurements were carried out using a Zahner CIMPS-X Photo-Electrochemical Workstation together with the THALES software package (Xpot-26366) on a photovoltaic cell of a two-electrode configuration with the metal oxide-based working electrode prepared according to the method above and a Pt counter electrode. The working and counter electrodes were spaced by a 0.13 mm paraffin spacer, and a small quantity of a redox electrolyte solution (an acetonitrile solution of KI (0.3 M) and I_2_ (0.05 M)) was injected between two electrodes through a small hole created on the Pt counter electrode. From the measurements of current-voltage (I-V) characteristics, the PCE was calculated based on an equation of PCE = *J*sc · *V*oc · FF/*I*ph, where *J*sc is the short-circuit current density, *V*oc is the open-circuit voltage, FF is the fill factor, and *I*ph is the source of stimulated light at 156 w/m^2^. Measurements were performed at least three times and the obtained numerical values were averaged. Electrochemical impedance spectra (EIS) were also acquired using the same instrument.

## 3. Results and Discussion

### 3.1. Optimization of Cascade FRET

The cascade FRET between fluorescence molecules of Py, AC, and AO in ethanol operates as a wavelength converter involving multi-step energy transfer. The primary donor (Py) is excited by high energy hν, and this energy transfers without emission of Py to the intermediate (AC), which serves as both donor and acceptor, transmitting its excitation energy to the final acceptor (AO). AO then emits energy, as illustrated in Figure 1. This emission occurs at a red-shifted wavelength. Thus, since sequential energy is transmitted across three chromophores with such a cascade energy transfer, the primary donor must possess the excitation spectrum that ideally fits the used light source, and the final acceptor provides the emission at the desired wavelength optimal for the visible range down to near-IR [15,16].

The UV-visible absorption, excitation, and emission spectra of Py, AC, and AO in an ethanol solution are shown in Figure 2. Py exhibited a strong emission band at 393 nm at a 335 nm exciting wavelength. The excitation spectrum emitted at 393 nm was similar to the absorption spectrum. The absorption spectrum of AC possessed a visible band at 432 nm. When the solution was excited at this wavelength, the corresponding emission band was observed at 478 nm. The excitation spectrum emitted by 478 nm also showed the same excitation band as the absorption band. AO showed an absorption band at 489 nm in the visible region. A pair of excitation and emission bands were observed at 489 and 520 nm, respectively [17,18,19]. The absorption/excitation and emission bands of C6 were observed at 450 and 504 nm, respectively.

A fluorescence emission band of a donor and an absorption spectrum of an acceptor should overlap so that the fluorescence donor material transfers the light energy to the acceptor. It can be noted from the observed absorption and emission bands that Py can play a donor due to the spectral overlaps of the Py-AC and Py-AO systems; AC can play an acceptor due to the spectral overlap of the Py-AC system, but it also behaves as a donor in the AC-AO system; AO plays an acceptor in the Py-AO and AC-AO systems. Thus, from Figure 3, which presents the combinations of an emission spectrum of an energy donor and a visible absorption spectrum of an energy acceptor, the Py-AC and AC-AO systems clearly indicate partial spectral overlaps so that they can be involved in energy transfer, although the Py-AO system would have poor energy transfer because of the lower spectral overlap [20]. Thus, AC is a good acceptor in the Py-AC system and a good donor in the AC-AO system and transfers energy from Py to the third dye, AO. As a result, the AC series can generate spectral overlap as the cascade FRET process from Py through AC to AO, including multiple individual energy transfers from Py to AC, AC to AO, and Py to AO. Conversely, with the C6 series, FRET acts on the Py-C6 and Py-AO systems, but it does not act between C6 and AO because the emission band of C6 is higher than the absorption band of AO.

Figure 4A shows fluorescence spectra of ethanol solutions of Py (3 µM) with AC and AO at different concentrations. The emission of Py decreased with increasing AC content and also slightly decreased with adding AO, although the emission intensity of both AC and AO increased. The photoenergy absorbed by Py is transferred to the acceptor AC and thus the emission of Py decreases. However, since FRET occurs slightly between Py and AO, the emission of Py decreases less in the Py-AO system. In ethanol solutions of AC with AO at different concentrations, as FRET occurred, the emission of AC decreased with increasing AO content. In the mixed ethanol solutions of Py (3 µM), AC (3 mM), and AO at various concentrations, the intensity of the emission band of Py was obviously weak and the emission of AC decreased with the addition of AO, indicating the energy transfer from donor Py to acceptor AC and from donor AC to acceptor AO—that is the cascade FRET [21,22,23,24].

The energy transfer between Py-C6, C6-AO, and Py-C6-AO was also examined (Figure 4B). The energy absorbed by Py was transmitted to C6 on the Py-C6 system. However, the emission band of C6 at 504 nm on the C6-AO system decreased relative to an emission band of AO (519 nm) with increasing concentration of AO, implying that there is no energy transfer between them, as estimated in Figure 3B. Thus, the emission spectra of the mixed ethanol solutions of Py (3 µM), C6 (3 mM), and AO at various concentrations displayed a decrease in a Py emission band, and the increase in an AO emission band overlapped with a C6 emission band. That is, FRET is limited between Py and C6.

An N719 dye molecule, a conventional light-harvesting dye on DSSC, has three absorption bands at 330, 384, and 526 nm, and these absorption bands of N719 overlap with emission bands of four fluorophores, Py, AC, C6, and AO, at 393, 478, 504, and 519 nm, respectively (Figure 5A). In fact, the emission spectra in Figure 5B indicate that the emission bands of the fluorophores at a constant concentration (3 mM) decreased after adding N719. This result indicates that N719 fulfills the role of an energy acceptor from Py, AC, C6, and AO, as energy donors, although a new emission band was not observed because N719 does not emit fluorescence, suggesting that the photon energies from the FRET dye are transferred to N719 and used for charge separation of N719.

### 3.2. Energy Transfer in Cascade FRET

Based on UV-visible absorption and fluorescence spectra, the energy transfer efficiency and related parameters in cascade FRET can be calculated using Equations (1)–(6) [25]. The energy transfer efficiency (*Φ*_Tx_) from a donor is given by:(1)$$\Phi_{\mathrm{Tx}}=1-\left(\frac{\Phi x}{\Phi_{\mathrm{Donor}}^{{}^{\circ}}}\right)$$
where *Φx* and $\Phi_{\mathrm{Donor}}^{0}$ are the quantum yields of the donor in the presence and absence of the acceptor, respectively. Then the quantum yield (*Φx*) is given by:(2)$$\Phi x=\Phi_{\mathrm{st}}\left(\frac{A_{\mathrm{st}}}{A_{x}}\right)\left(\frac{F_{x}}{F_{\mathrm{st}}}\right){\left(\frac{n_{x}}{n_{\mathrm{st}}}\right)}^{2}$$
where *A* is the absorbance of an absorption band, *F* is the area intensity of an emission band, and *n* is the refractive index of the solvent. The subscripts st and x indicate the standard (Py) and cascade dyes, respectively. Using the observed absorption and emission spectra (Figure 2) and the known quantum yield (0.65) of Py [26], the quantum yield of AC was calculated to be 0.06 based on Equation (2). This value was rather low as a coumarin derivative because the carbonyl group could suppress the fluorescence [27,28].

The distance (*r*_x_) between the donor and acceptor is given by:(3)$$r_{x}={\left(\frac{1}{\Phi_{\mathrm{Tx}}}-1\right)}^{1/6}R_{0x}$$
where *R*_0x_ is the Förster critical distance where the resonance energy transfer is 50% efficient [29]. The Förster critical distance between the donor and acceptor is given by:(4)R0x=0.211κ2n−4Φst,x Jxλ16
where *κ*^2^ is the orientation factor, *n* is the refractive index of the medium, and *Φ*_st,x_ is the quantum yield of the donor. The orientation factor is assumed to be equal to 2/3, which is appropriate for dynamically random averaging of donor and acceptor [30]. The parameter *J*_x_(λ) expresses the degree of spectral overlap between an emission band of the donor and a UV-visible absorption band of the acceptor (see Figure 3) [31], and it is given by:(5)$$J_{x}\left(\lambda \right)=\frac{\int F_{\mathrm{Donor}}\left(\lambda \right)\varepsilon \left(\lambda \right)\lambda^{4}d\lambda }{\int F_{\mathrm{Donor}}\left(\lambda \right)d\lambda }$$
where *F*_Donor_(λ) in arbitrary units is the emission intensity of the donor at a wavelength λ in nm, and ε(λ) is the molar coefficient of the acceptor in M^−1^cm^−1^. The spectral overlap integrals [*J*_x_(λ)] were obtained from Figure 3, and *J*_x_(λ) calculated using Equation (5) was 5.986×10^14^ M^−1^cm^−1^(nm)^4^ for the Py–AC system and 17.28 × 10^14^ M^−1^cm^−1^(nm)^4^ for the AC-AO system. However, it was very small for the Py-AO system. The Förster critical distance (*R*_0x_) calculated from Equation (4) was 43.4 Å between Py and AC and 39.0 Å between Py and C6 [32]. Because the FRET depends on the resonance between the electron clouds of the donor and acceptor, it is reasonable that the similarity in molecular structures of AC and C6 results in similar *R*_0x_ values. The Förster critical distance (*R*_0x_) was calculated to be 34.8 Å between AC and AO and 28.54 Å between Py and AO [33]. The rate constant (*k*_Tx_) for energy transfer between donor and acceptor is given by:(6)$$k_{\mathrm{Tx}}=\frac{1}{\tau_{\mathrm{Donor}}^{0}}{\left[\frac{R_{0x}}{r_{x}}\right]}^{6}$$
where $\tau_{\mathrm{Donor}}^{0}$ (=410 ns [26]) is the excited state lifetime of the donor in absence of energy transfer.

Concerning the dye concentration dependence on parameters relating to Py-AC-AO cascade FRET, spectra of solutions with different dye concentrations using three mother dye solution series listed in Appendix A were measured, and the calculated parameters are listed in Appendix A and plotted in Figure 6. When the energy transfer efficiency (*Φ*_T_) was plotted as a function of AC concentration (Figure 6A), the efficiency increased with increasing AC concentration in the Py-AC FRET system and reached 0.90–0.95. If AO was added into the Py-AC solution, it increased further to 0.95–0.97. The distance (*r*_1_) between the donor (Py) and acceptor (AC) was plotted in Figure 6B as a function of AC concentration. The distance decreased with increasing AC concentration and further addition of AO in the Py-AC solution decreased the distance further.

Equation (3) can be rewritten to Equation (7):(7)$$\Phi_{Tx}=\frac{R_{0x}^{6}}{R_{0x}^{6}+r_{x}^{6}}\;\mathrm{and}\;1/\Phi_{Tx}=1+r_{x}^{6}/R_{0x}^{6}$$

Equation (7) indicates that the inverse energy transfer efficiency (*Φ*_Tx_) is dependent on the distance *r*_x_.^33^ Figure 6C shows that the energy transfer efficiency increased almost linearly with decreasing *r*_x_/*R*_0x_ but the efficiency deviated from linearity at an *r*_x_/*R*_0x_ of less than 0.7. From this deviation of the curve, it can be inferred that (1) an effective Py-AC pair is formed at high *r*_x_/*R*_0x_ ratios, which is to say, low dye concentrations, and (2) when *r*_x_/*R*_0x_ ratio decreases to less than 0.7, the number of effective Py-AC pairs cannot continue to increase, since the molecular number of Py is limited, and, at the same time, the emission of Py is quenched easily with the addition of AC. Because of these two reasons, the increase in the energy transfer efficiency will slow down and approach constant efficiency (1.0) [29]. When *r*_x_/*R*_0x_ is equal to one, the energy transfer efficiency reaches 0.5 (50%). This obeys the definition of *R*_0_, which is the Förster critical distance between the donor and acceptor described above [34].

The relation of the rate constant (*k*_T_) with the distance (*r_x_*) between Py and AC is plotted in Figure 6Da. When the distance becomes short, the rate constant rapidly increases, as supported by Equation (6). The rate constant also relates to the energy transfer efficiency (*Φ_T_*) in relation to Equation (7). It can be referred that the energy transfer efficiency increases when the rate constant increases with decreasing distance *r*_1_ (Figure 6Db). According to the above discussion, if the concentrations of AC and AO increase, the distance, *r*_1_ between Py and AC is reduced. At the same time, the rate constant (*k*_T_) and the energy transfer efficiency (*Φ_T_*) of Py increase. During this process, the quantum yield of Py decreases, since energy transfer occurs. Therefore, Py, AC, and AO can be involved in cascade FRET.

As the major energy transfer efficiency (*Φ*_Tx_) occurs between the Py donor and AC acceptor, the quenching mechanism between two fluorescence molecules can be depicted using the Stern–Volmer law used for bimolecular processes [35]. There are three types of quenching equations: static, dynamic, and combined static-dynamic Stern–Volmer types as described in Equations (8)–(10), respectively [4].
(8)$$\frac{\Phi_{\mathrm{st}}}{\Phi }=\frac{I_{0}}{I}=1+\tau_{\mathrm{Donor}}^{0}k_{q}\left[Q\right]=1+K_{\mathrm{SV}}\left[Q\right]\;(K_{\mathrm{SV}}=\tau_{\mathrm{Donor}}^{0}\;k_{q})$$
(9)$$\frac{\Phi_{\mathrm{st}}}{\Phi }=\frac{I_{0}}{I}=1+K\left[Q\right]$$
(10)$$\frac{\Phi_{\mathrm{st}}}{\Phi }=\frac{I_{0}}{I}=\left(1+K_{\mathrm{SV}}\left[Q\right]\right)\left(1+K\left[Q\right]\right)$$
where *I*_0_ and *I* are the emission intensities of the donor in the absence and presence of an acceptor, respectively, *k*_q_ is the bimolecular quenching rate constant, $\tau_{\mathrm{Donor}}^{0}$ is the lifetime of the donor in absence of an acceptor; *K* is the association constant of the ground state complex [36] and [*Q*] is the concentration of a quencher (acceptor). According to these three equations, Equations (8) and (9) must be straight lines in plots of *Φ*_st_/*Φ* against the concentration of quencher, while a plot of a combined static-dynamic Stern–Volmer equation should show upward curvature. The ratio of *I*_3,0_/*I*_3_ for the third emission band of Py was then plotted as a function of the concentration of AC. Since the obtained plots were fitted with a second-order polynomial (see Figure 6E), the quenching behavior was assumed to follow a combined static-dynamic Stern–Volmer equation. The rapid increase in *I*_3,0_/*I*_3_ indicates that the emission from the Py fluorophore can be quenched easily by an AC quencher [37].

The quenching factors are classified into two mechanisms as follows. In static quenching, since a nonfluorescent complex consisting of quencher (AC)-fluorophore (Py) is formed in the ground state, the photoluminescence property of the fluorophore (Py) will disappear [36]. Static quenching easily occurs even at high concentrations because the number of quencher (AC)-fluorophore (Py) complexes will rise. In dynamic quenching, the quencher (AC) collides with the excited fluorophore (Py). This will allow the fluorophore to return to the ground state before emitting fluorescence. Therefore, energy is lost during the energy transfer process [38]. According to the analysis described above, even though the concentrations of the mother solution (I-III) are different, the three curves are close or overlap completely, as seen in Figure 6. This indicates that the energy transfer efficiency and the rate constant are controlled by the distance between the donor and acceptor and not their individual concentrations [39].

### 3.3. Enhancement of Light Harvesting on DSSC Systems Equipped Cascade FRET

To study the effect of cascade FRET dopants for DSSCs composed of photovoltaic electrodes (ZnO@Cdot, ZnO@N719, and ZnO@Cdot@N719) and adsorbed FRET dyes (Py, Py-AC, and Py-AC-AO), I-V curves were measured, as shown in Figure 7. The current varied depending on the electrodes and the addition of FRET dyes, although the voltage was similar at around 8 V. These variations were reflected in the parameters that were calculated based on the I-V curves, as listed in Table 1. PCE values of ZnO@Cdot and ZnO@N719 were very low but the coexistence of Cdot with N719 synergistically increased PCE. Moreover, the addition of FRET dyes on ZnO@N719 and ZnO@Cdot@N719 further influenced PCE. In particular, the increase in the short circuit current density (Jsc) directly affected on the increase in PCE after adding FRET dyes, although the open circuit voltage (Voc) and the Fill factor (FF) values did not show much dependence on the additives. The sequential addition of Py, AC, and AO promoted PCE values, step-by-step, and the three FRET dyes (16.6 wt%)-adsorbed ZnO@Cdot@N719 DSSC exhibited the highest PCE (10.37 + 0.12%), which was almost double that of ZnO@Cdot@N719 DSSC without FRET dyes. This increased efficiency indicates the energy transfer between the FRET dyes and N719. Such charge transfers were clarified from the variation of their emission spectra (Figure 5) as described above, and these photovoltaic results demonstrate increased light-harvesting characteristics in the DSSC. The photovoltaic performances of the DSSCs were significantly enhanced with the addition of the energy-transferable fluorescence materials on the ZnO@Cdot@N719 electrode due to their light-harvesting characteristics.

The effect of fluorophores (Py, C6, and AO) shows a similar tendency to the combination of Py, AC, and AO, but the numerical values of the current (Jsc) and, thus, PCE were lower than those of the combination of Py, AC, and AO, as seen in Figure 7B and Table 1. These results confirm that the effect of the fluorophores on the performance of the DSSCs depends on the energy transfer from the fluorophores to the acceptors, and the energy transfer has less effect on the combination of Py, C6, and AO than the combination of Py, AC, and AO. Figure 7A,B show I-V curves of DSSCs at three concentrations (14.3 wt%, 16.7 wt%, and 20.0 wt%) of cascade FRET dyes. When the concentration increased, the current changed, or the voltage varied. The results indicate that the maximum performance occurred at 16.7 wt% [40,41].

The photovoltaic J–V characteristics of NiO@N719 and NiO@Cdot@N719 photoanode p-type DSSCs sensitized by FRET dyes were measured, and the results are shown in Figure 7C and Table 1. A conspicuous advantage of the NiO@Cdot@N719 electrodes compared to NiO@N719 electrodes was the high photocurrent and photovoltage. The charge transporting ability is reflected more on NiO@Cdot@N719 than NiO@N719, as indicated by the photocurrent. The short-circuit photocurrent and the open-circuit voltage of NiO@Cdot@N719 DSSCs increased about three times and about 40%, respectively, compared to those of NiO@N719 DSSCs, emphasizing the importance of Cdot in the charge separation and adsorption of dyes [12]. Superiority of the I-V curves of NiO-based electrodes against those of ZnO-based electrodes were also found in the form of higher current and lower voltage. It is known that p-type semiconductors exhibit a low hole transfer rate despite a large optical band gap and high ionization potential and, thus, have a poorer performance than n-type semiconductors. However, the high photovoltaic performance of NiO@Cdot@N719 is due to the enhanced effective surface area (61.27 m^2^/g) of NiO@Cdot compared to ZiO@Cdot (38.76 m^2^/g) [9,12]. Cdot adsorbed on the surface of semiconductor particles can play the role of photosensitizer, improve the electron transport and ionic motion and enlarge the contact area between the electrode and electrolyte. Thus, the NiO@Cdot composite exhibits a higher conductivity and suppression of recombination than ZnO@Cdot [10], and the same situation happens even with the cascade FRET DSSCs [42]. In particular, the improvement in the photocurrent originates more likely from the result of enhanced light harvesting in addition to charge transfer ability. Thus, PCE values for cascade FRET NiO photoanode DSSCs were higher than those for cascade FRET ZnO photoanode DSSCs. These multistep cascade charge transfer processes displayed remarkably higher photovoltaic efficiencies compared to cells without cascade FRET dyes. Thus, cascade FRET dyes play the role of electron donors and acceptors, and hole injection is accompanied by an ultrafast electron shift reaction that leads to an extremely long-lived charge-separated state [43]. Figure 8 illustrates the processes by which the photoenergy excited by Py is transferred to N719 through the cascade FRET and the highly charge-separated N719 enhances the efficiency of semiconductor@Cdot DSSCs.

## 4. Conclusions

The addition of fluorophore materials as a photosensitizer resolves two problems associated with DSSCs: the utilization at a wide solar spectrum range and the interruption of the recombination process. When Py is introduced to a ZnO@Cdot@N719 DSSC, photons are absorbed by Py upon illumination. If acceptor fluorescence dyes (AC and AO) are added in the photoanode, the energy excited by Py is transferred to them, and, then, to N719, and finally to the semiconductors after the charge separation. Such processes were confirmed in the present investigation in which cascade energy transfer based on the FRET principle was designed and the photon energy of a donor excited by absorbing UV light at 335 nm was transferred to the neighboring first acceptor fluorescence molecule, followed by transfer to the second acceptor, and emission at 519 nm from the second acceptor. When the cascade FRET dyes were incorporated in DSSCs, the cascade FRET effect was able to improve the light-harvesting in N719 and could also enhance the efficiency of DSSC, and at the same time, the recombination between electron and hole was reduced due to their light-harvesting characteristics and the electron lifetime effect, as illustrated in Figure 8. As a result, an improved PCE (10.5%) of the ZnO-based DSSC was achieved based on the cascade FRET process at 156 W/m^2^ illumination. Moreover, this study provided further improved performance in NiO-based p-type DSSCs because the enhancement was higher than in ZnO-based n-type DSSCs.

Literature reported FRET-including DSSCs are rather few. Reports were focused on co-sensitization and FRET of TiO_2_ DSSCs, and PCE values of these systems were at most 5.17% [44,45,46,47,48]. Compared with the present report, these PCE values are very low. The difference between our system and these systems is that our system consists of a cascade FRET and additional co-photosensitizers (N719 and Cdot). These multi-photosensitizer system effects to enhance PCE and, moreover, Cdot plays a role in intensifying the adsorption of the dyes and discouraging the recombination of charges on the semiconductors [12]. Consequently, the present FRET system is effective as a photovoltaic device because of the additional enhancement effects of the multi-photosensitizer system and Cdot. This research confirms that the adsorption of multiple organic dyes harvesting sunlight and preventing the recombination effect on the performance of DSSCs should be valuable for the advancement of DSSCs and must be taken into account in future research.

## Figures and Tables

**Figure 1 nanomaterials-12-04085-f001:**
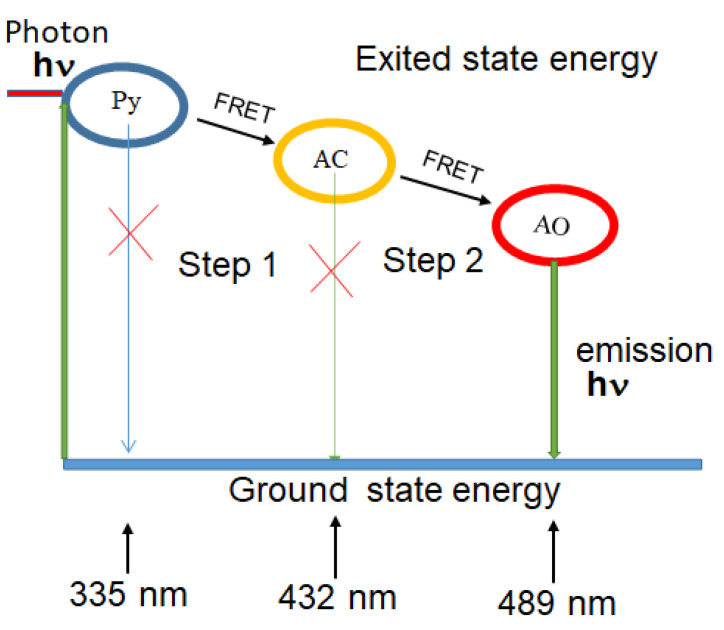
A schematic diagram of the cascade FRET process from donor molecule (Py) through first acceptor molecule (AC) to second acceptor molecule (AO). Wavelengths represent absorption bands.

**Figure 2 nanomaterials-12-04085-f002:**
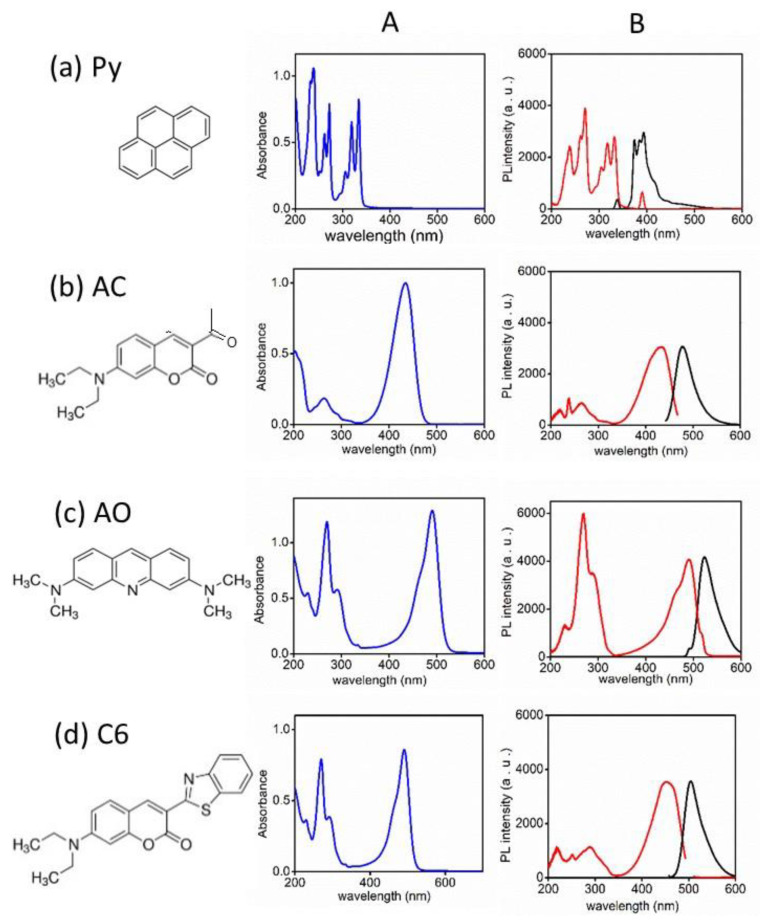
(**A**) UV-visible absorption spectra and (**B**) fluorescence excitation and emission spectra. (**a**) Py, (**b**) AC, (**c**) AO, and (**d**) C6. The concentration of each solution is 10 μM.

**Figure 3 nanomaterials-12-04085-f003:**
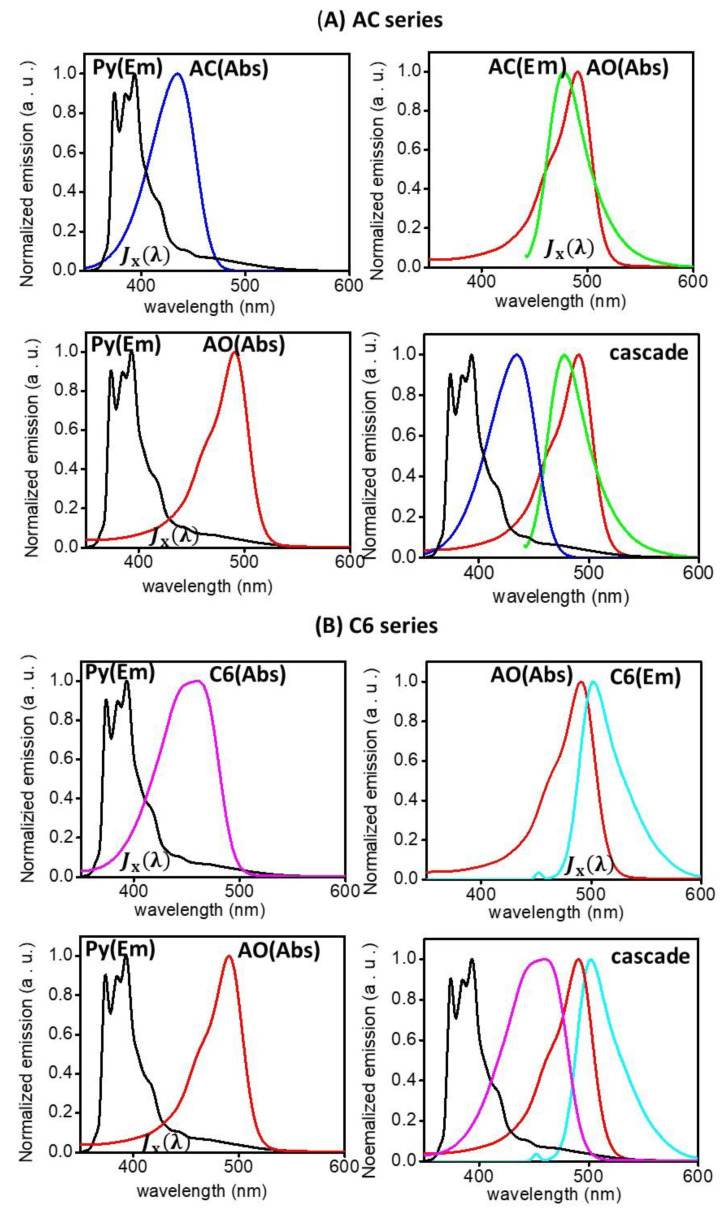
The emission and visible absorption spectral overlaps of (**A**) Py(Em)-AC(Abs), AC(Em)-AO(Abs), Py(Em)-AO(Abs) and cascade FRET process and (**B**) Py(Em)-C6(Abs), C6(Em)-AO(Abs), Py(Em)-AO(Abs) and cascade FRET process. *J_x_*(λ) indicates the degree of spectral overlap between the emission band of a donor and the UV-visible absorption band of an acceptor.

**Figure 4 nanomaterials-12-04085-f004:**
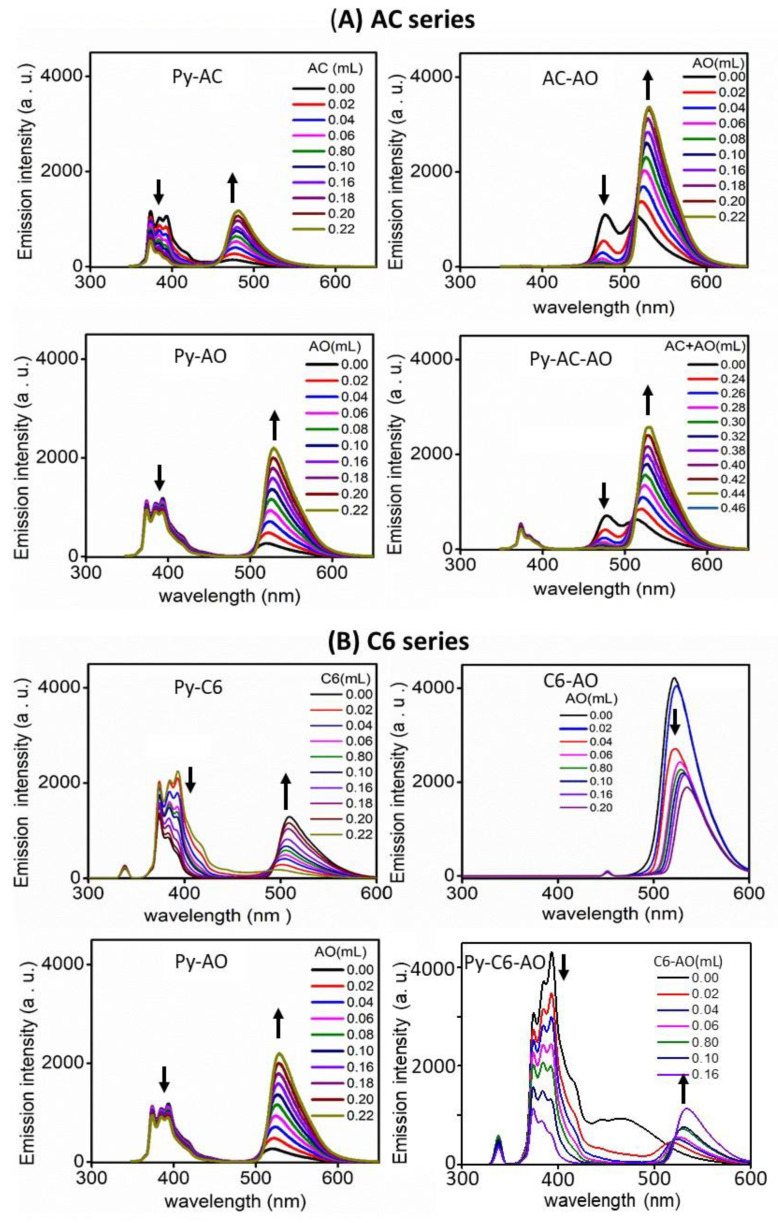
Emission spectra of ethanol solutions of (**A**) Py(3 μM)-AC, Py(3 μM)-AO, AC(3 mM)-AO and Py(3 μM)-AC(3 mM)-AO and (**B**) Py(3 μM)-C6, Py(3 μM)-AO, C6(3 mM)-AO and Py(3 μM)-C6(3 mM)-AO.

**Figure 5 nanomaterials-12-04085-f005:**
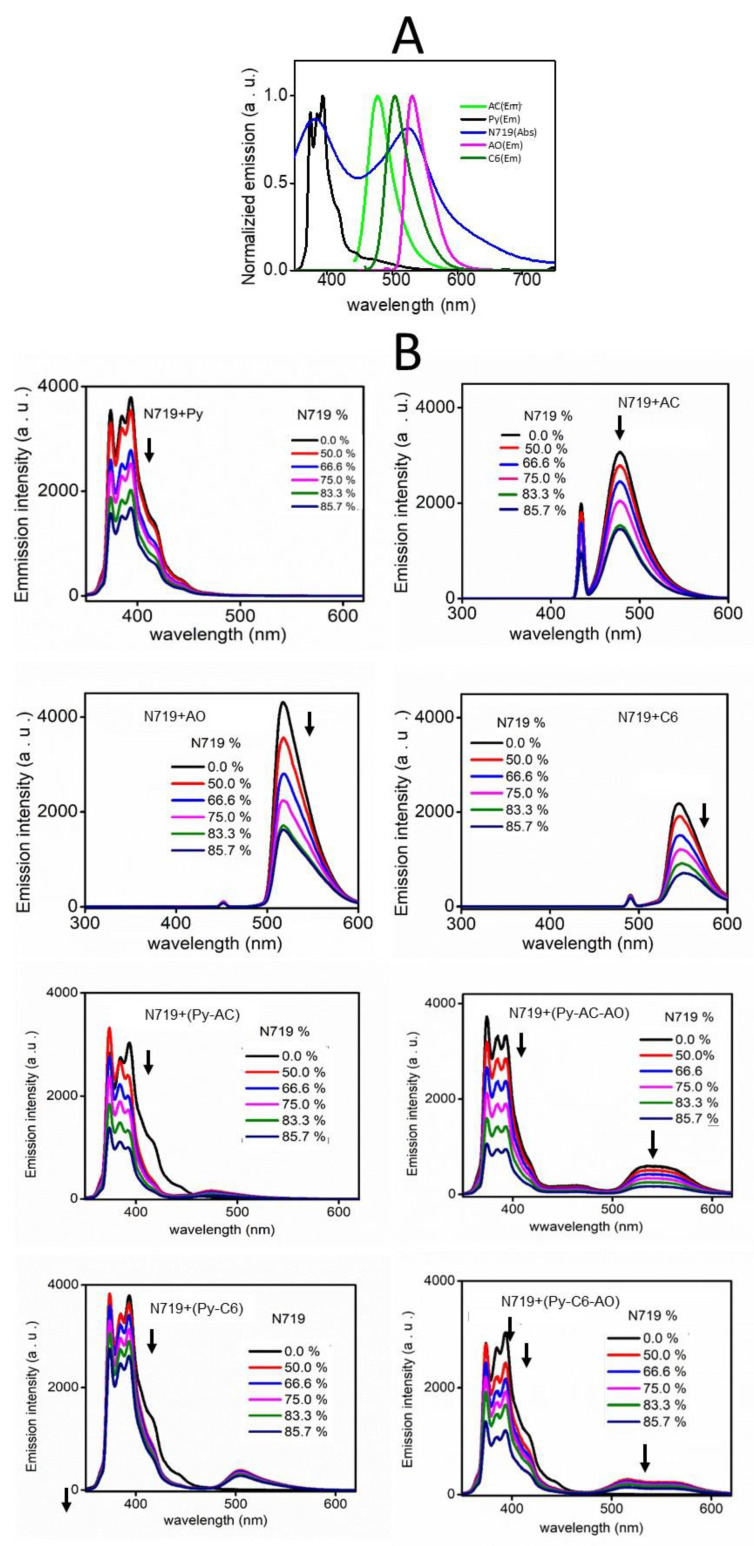
(**A**) Normalized emission spectra of FRET fluorescent dyes and an absorption spectrum of N719 (0.1 mM in ethanol). (**B**) Fluorescence emission spectra of ethanol solutions of Py(3 mM), AC(3 mM), AO(3 mM), C6(3 mM), Py(3 mM)-AC(3 mM), Py(3 mM)-AC(3 mM)-AO(3 mM), Py(3 mM)-C6(3 mM), and Py(3 mM)-C6(3 mM)-AO(3 mM) mixed with N719 at various concentrations.

**Figure 6 nanomaterials-12-04085-f006:**
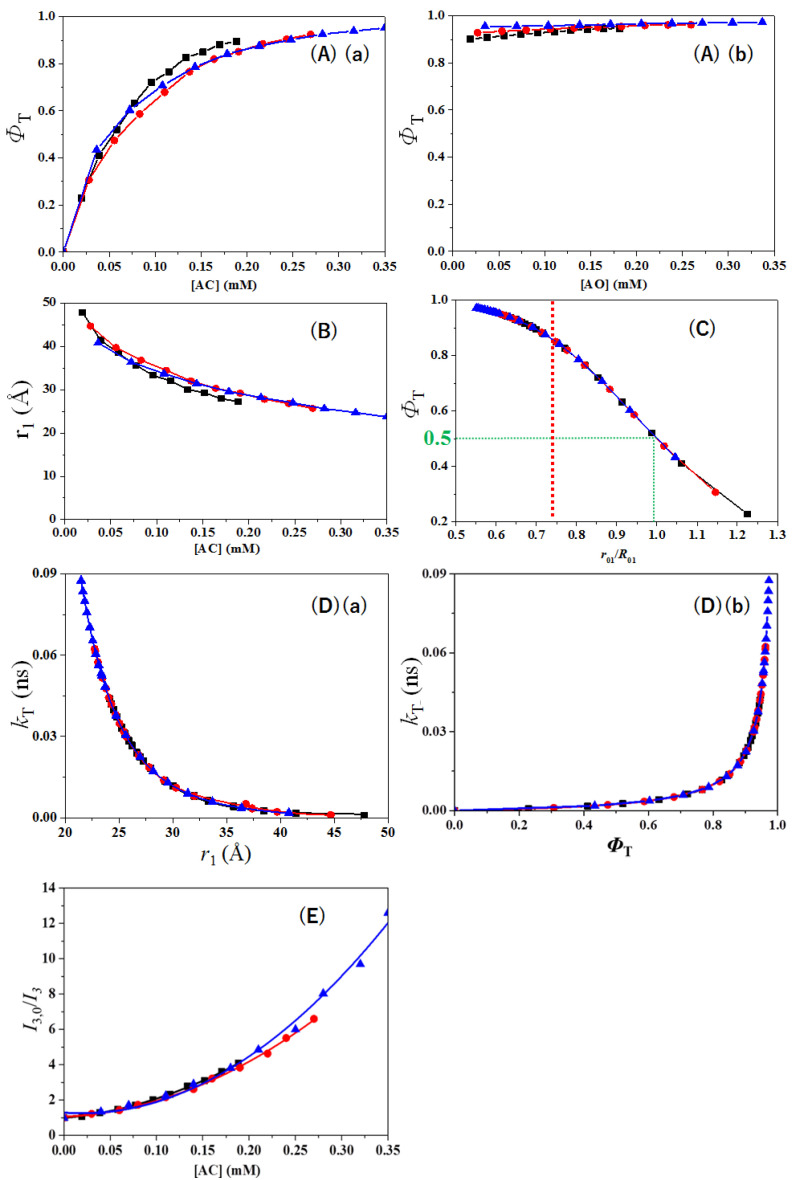
(**A**) Energy transfer efficiency of Py versus concentration of (**a**) AC and (**b**) AO. (**B**) Distance between Py and AC versus concentration of AC. (**C**) Energy transfer efficiency of Py versus r_1_/R_01_. (**D**) Rate constant of Py versus (**a**) distance between Py and AC and (**b**) energy transfer efficiency of Py. (**E**) I_3,0_/I_3_ for the third emission band of Py versus concentration of AC. Mother solution: 
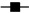
: Py(3.5 μM)-AC(4.9 mM)-AO(4.9 mM), 
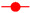
: Py(5 μM)-AC(7 mM)-AO(7 mM), and 
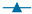
: Py(6.5 μM)-AC(9.1 mM)-AO (9.1 mM).

**Figure 7 nanomaterials-12-04085-f007:**
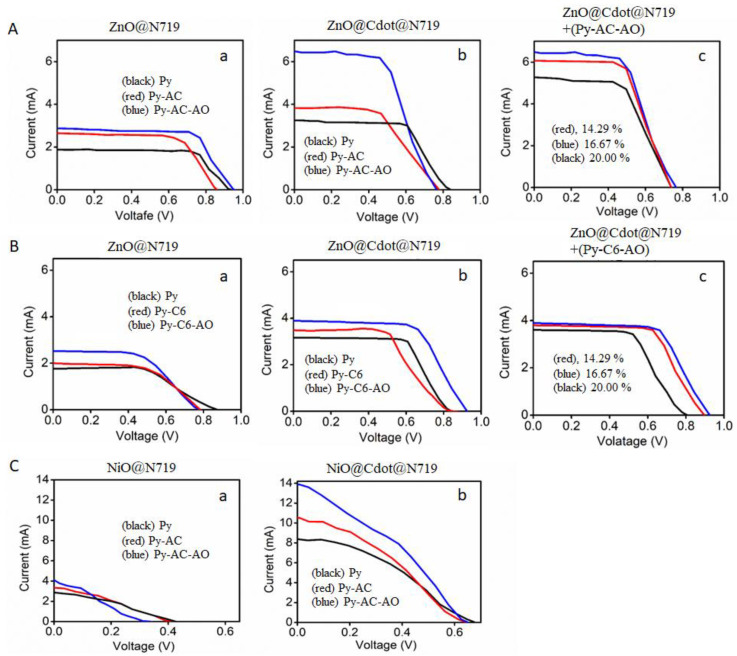
I–V curves of cascade FRET DSSCs. (**A**) (**a**) ZnO@N719 and (**b**) ZnO@Cdot@N719 under addition of FRET dyes; (black) Py, (red) Py-AC, and (blue) Py-AC-AO. (**c**) ZnO@Cdot@N719 + (Py-AC-AO) at different FRET dye concentrations; red: 14.3wt%, blue: 16.7wt%, black: 20.0wt%. (**B**) (**a**) ZnO@N719 and (**b**) ZnO@Cdot@N719; (black) Py, (red) Py+C6, (blue) Py+C6+AO. (**c**) ZnO@Cdot@N719 + (Py-C6-AO); red: 14.3wt%, blue: 16.7wt%, black: 20.0wt%. (**C**) (**a**) NiO@N719 and (**b**) NiO@Cdot@N719 under addition of FRET dyes; (black) Py, (red) Py-C6 and (blue) Py-C6-AO.

**Figure 8 nanomaterials-12-04085-f008:**
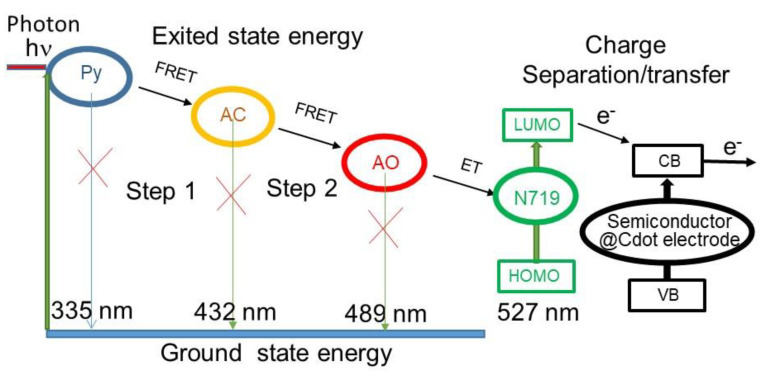
Schematic diagram of energy transfer from cascade FRET fluorophores to N719 and charge separation/transfer in semiconductor@Cdot@N719 DSSC.

**Table 1 nanomaterials-12-04085-t001:** Photovoltaic parameters of cascade FRET DSSCs.

Electrode Material	J_SC_ (mA/cm^2^)	V_oc_ (V)	FF	PCE (%)
ZnO@Cdot	0.006 ± 0.001	0.17 ± 0.10	0.20 ± 0.01	0.001 ± 0.000
ZnO@Cdot+(Py-AC-AO)	1.30 ± 0.01	0.15 ± 0.03	0.17 ± 0.02	0.21 ± 0.01
ZnO@N719	0.61 ± 0.10	0.32 ± 0.02	0.65 ± 0.01	0.80 ± 0.01
ZnO@N719+Py	1.73 ± 0.07	0.83 ± 0.06	0.31 ± 0.01	2.80 ± 0.18
ZnO@N719+(Py-AC)	2.61 ± 0.01	0.95 ± 0.05	0.28 ± 0.02	4.45 ± 0.15
ZnO@N719+(Py-AC-AO)	2.85 ± 0.02	0.95 ± 0.11	0.29 ± 0.02	4.61 ± 0.20
ZnO@Cdot@N719	2.34 ± 0.03	0.67 ± 0.10	0.59 ± 0.05	5.92 + 0.01
ZnO@Cdot@N719+Py	3.25 ± 0.09	0.81+0.14	0.37 ± 0.11	6.24 + 0.13
ZnO@Cdot@N719+(Py-AC)	3.82 ± 0.03	0.79 ± 0.12	0.38 ± 0.02	7.31 ± 0.10
ZnO@Cdot@N719+(Py-AC-AO(14.3wt%))	6.10 ± 0.03	0.77 ± 0.30	0.30 ± 0.05	9.03 ± 0.14
ZnO@Cdot@N719+(Py-AC-AO(16.7wt%))	6.44 ± 0.01	0.79 ± 0.20	0.32 ± 0.03	10.37 ± 0.12
ZnO@Cdot@N719+(Py-AC-AO(20.0wt%))	5.40 ± 0.04	0.77 + 0.11	0.30 ± 0.11	7.99 ± 0.13
ZnO@N719+Py+C6	2.00 ± 0.01	0.78 ± 0.07	0.31 ± 0.09	3.10 ± 0.33
ZnO@N719+Py+(C6-AO)	2.50 ± 0.03	0.79 ± 0.21	0.30 ± 0.02	3.79 ± 0.23
ZnO@Cdot@N719+Py	3.25 ± 0.09	0.81 ± 0.14	0.37 ± 0.11	6.24 ± 0.13
ZnO@Cdot@N719+(Py-C6)	3.50 ± 0.11	0.83 ± 0.22	0.38 ± 0.02	7.01 ± 0.21
ZnO@Cdot@N719+(Py-C6-AO(14.3wt%))	3.81 ± 0.02	0.92 ± 0.12	0.37+0.02	8.35 ± 0.14
ZnO@Cdot@N719+(Py-C6-AO(16.7wt%))	3.94 ± 0.11	0.93 ± 0.20	0.41 ± 0.01	9.63 ± 0.11
ZnO@Cdot@N719+(Py-C6-AO(20.0wt%))	3.61 ± 0.11	0.81 ± 0.01	0.36 ± 0.02	6.74 ± 0.22
NiO@N719+Py	2.85 ± 0.02	0.45 ± 0.10	0.22 ± 0.03	1.84 ± 0.10
NiO@N719+(Py-C6)	3.42 ± 0.01	0.45 ± 0.15	0.23 ± 0.06	2.25 ± 0.22
NiO@N719+(Py-C6-AO)	4.10 ± 0.05	0.38 ± 0.32	0.26 ± 0.02	2.59 ± 0.24
NiO@Cdot@N719+Py	8.13 ± 0.03	0.65 ± 0.13	0.20 ± 0.01	6.75 ± 0.15
NiO@Cdot@N719+(Py-C6)	10.31 ± 0.01	0.62 ± 0.20	0.21 ± 0.04	8.34 ± 0.10
NiO@Cdot@N719+(Py-C6-AO)	13.95 ± 0.05	0.62 ± 0.16	0.22 ± 0.02	11.36 ± 0.25

## Data Availability

Not applicable.

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
