# Peer review of "Cascade Förster Resonance Energy Transfer Studies for Enhancement of Light Harvesting on Dye-Sensitized Solar Cells"

_nanomaterials, 2022, doi:10.3390/nano12224085_

Round 1

Reviewer 1 Report

This manuscript describes data-rich and scientifically reliable work. I basically recommend publication of this work in Nanomaterials. Although it is scientifically well conducted, the manuscript constructions (especially figures) are not in sufficiently high level. I want to suggest several revisions for further improvements.

1) Generally position of equations in lines looks strange. Is it OK?

2) Relation between physical phenomena and actual materials (DSSC constructions) is not so clear in the current figure set. Figure 1 had better include chemical structures and their relation of DSSC components.

3) In the Table 1, some error representations are wrong. Minimum orders of main values and error values have to be same.

3) Addition of one figure to explain conclusive mechanisms may be helpful.

Author Response

Added a file.

Reviewer 2 Report

The paper titled Cascade Förster Resonance Energy Transfer Studies for Enhancement of Light Harvesting on Dye-Sensitized Solar Cells by Mulugeta Tesema Efa et al is quite an interesting paper exploiting relevant potential properties of DSSC that may impact the future. However, mandatory revision is required prior to taking any decision about its acceptance, aiming to clarify certain points and make it more attractive for a broad community. See the comments below.

Title: It is aligned with the study performed

Abstract: It is missing key performance indicators of the study performed as well as what are the contributions to the advance of the present state of the art.

Introduction: When addressing the topic, as mentioned in the abstract, please refer to the role that n-type (ZnO) and p-type (NiO) materials will have on the transport of carriers within the dye-sensitized solar cells (DSSCs). Please see for instance Panigrahi, S et al in  Light-induced current mapping in oxide-based solar cells with nanoscale resolution, Mar 2018 | SOLAR ENERGY MATERIALS AND SOLAR CELLS, 176 , pp.310-317 and Nandy, et al in Current transport mechanism at metal-semiconductor nanoscale interfaces based on ultrahigh density arrays of p-type NiO nano-pillars, 2013 | NANOSCALE, 5 (23), pp.11699-11709.

When referring to the energy transfer mechanism worth also identifying the role that interfaces will have.

Experimental procedures: How many samples were processed? Can you identify the size and architecture of the structures processed? How reproducible ad reliable the process and the structures are? What is the error associated when evaluating structures processed in the same batch but in different spatial locations? What are the errors associated to structures performances from batch to batch? What are the environmental conditions in which the structures were tested?

Results and Discussion: Well done. Did you find any type of persistent photoconductivity effect in the structures evaluated that may affect the transport of carriers? Can you better explain the role of the interfaces on the set of performances achieved, namely concerning the recombination mechanism? Do you think that we may have geminate recombination?

Conclusions: It is missing a summary of quantified key performance indicators of the study performed.

References: Need to be updated

Figures: Very good

Tables: OK

Author Response

reply to reviewer 2 docx is correct.

Round 2

Reviewer 2 Report

The was properly revised and I agree with the st of changes performed